# An Update on Nucleolar Stress: The Transcriptional Control of Autophagy

**DOI:** 10.3390/cells12162071

**Published:** 2023-08-15

**Authors:** Astrid S. Pfister

**Affiliations:** Institute of Biochemistry and Molecular Biology, Ulm University, 89081 Ulm, Germany; astrid.pfister@uni-ulm.de; Tel.: +49-731-50023390

**Keywords:** autophagy, nucleolar stress, ribosome biogenesis, signaling, transcription factor

## Abstract

Nucleolar stress reflects a misfunction of the nucleolus caused by a failure in ribosome biogenesis and defective nucleolar architecture. Various causes have been reported, most commonly mutation of ribosomal proteins and ribosome processing factors, as well as interference with these processes by intracellular or ectopic stress, such as RNA polymerase I inhibition, ROS, UV and others. The nucleolus represents the place for ribosome biogenesis and serves as a crucial hub in the cellular stress response. It has been shown to stimulate multiple downstream consequences, interfering with cell growth and survival. Nucleolar stress induction is most classically known to stimulate p53-dependent cell cycle arrest and apoptosis. Nucleolar stress represents a friend and enemy at the same time: From a pathophysiological perspective, inactivation of the nucleolar function by mutation or stress conditions is connected to multiple diseases, such as neurodegeneration, cancer and ribosomopathy syndromes. However, triggering the nucleolar stress response via specific chemotherapeutics, which interfere with nucleolar function, has beneficial effects for anti-cancer therapy. Interestingly, since the nucleolar stress response also triggers p53-independent mechanisms, it possesses the potential to specifically target p53-mutated tumors, which reflects the most common aberration in human cancer. More recent data have shown that the nucleolar stress response can activate autophagy and diverse signaling cascades that might allow initial pro-survival mechanisms. Nevertheless, it depends on the situation whether the cells undergo autophagy-mediated apoptosis or survive, as seen for autophagy-dependent drug resistance of chemotherapy-exposed tumor cells. Given the relatively young age of the research field, precise mechanisms that underly the involvement of autophagy in nucleolar stress are still under investigation. This review gives an update on the emerging contribution of nucleolar stress in the regulation of autophagy at a transcriptional level. It also appears that in autophagy p53-dependent as well as -independent responses are induced. Those could be exploited in future therapies against diseases connected to nucleolar stress.

## 1. Background

### 1.1. Ribosome Biogenesis and Nucleolar Stress

Ribosomes represent RNA/protein complexes that function as crucial ribozymes for translation and, thus, cellular growth and survival. The process of building ribosomes is highly complex and has to be tightly regulated. It is orchestrated by the huge ribosome biogenesis machinery and is located at specific sub-nuclear compartments termed nucleoli (Figure 1) [1,2]. 

Eukaryotic 80S ribosomes are formed by the assembly of the large 60S and small 40S ribosomal subunits, which consist of a combination of rRNAs together with multiple ribosomal proteins. Ribosome biogenesis requires the transcription of an initial polycistronic rRNA precursor. This *47S* precursor is transcribed by the action of RNA polymerase I (RNA POL I). Afterward, the *47S* transcript undergoes several cleavage and chemical modification steps with the help of a multitude of ribosome processing factors to finally form the *5.8S*, *18S* and *28S* rRNAs [1,2]. In contrast, the *5S* rRNA is transcribed by RNA POL III. The large 60S subunit contains the matured *5S*, *5.8S* and *28S* rRNAs and the small 40S ribosomal subunit the *18S* rRNA. Together with ribosomal proteins, they build mature ribosomes. The process of ribosome biogenesis is regulated by key signaling cascades. For instance, mTOR (mammalian Target of rapamycin) signaling affects the production of ribosomes at multiple levels, including rRNA transcription and protein synthesis [3]. Additionally, active Wnt signaling drives ribosome biogenesis through direct and indirect mechanisms [4]. Wnt drives the expression of *CMYC* that in turn activates RNA POLs I–III. In addition, Wnt signaling activates the expression of ribosome biogenesis factors that are essential for the maturation steps (Figure 1) [4]. 

Nucleoli are highly dynamic structures that assemble around rDNA clusters in a cell cycle-dependent manner during the G_1_ phase when Wnt signaling is active [4,5]. Strikingly, the nucleolus has been noticed as a platform for sensing and reacting to cellular stress by phenotypical and functional responses [6,7]. The term nucleolar stress is used in conditions of failure in ribosome biogenesis, which can go together with nucleolar disruption. As nucleolar size and morphology are coupled to nucleolar function, key nucleolar factors such as NPM (Nucleophosmin), Fibrillarin or UBF-1 (Upstream-binding factor 1) are used as markers to visualize the changes occurring during nucleolar stress [8]. In response to stress, nucleolar factors are relocalized to the nucleolar periphery or the nucleoplasm (Figure 2). Nucleolar stress goes together with an oxidized state of nucleoli, where NPM dissociates from nucleolar nucleic acids and translocates to the nucleoplasm during the process of nucleolar segregation [9]. 

Of note, nucleolar stress can be triggered by multiple conditions interfering with nucleolar integrity: Mutation of ribosome biogenesis factors or processing factors, intracellular stress, such as ROS (reactive oxygen species) and extrinsic stress, such as UV irradiation or chemotherapeutic drugs. Activation of the classical nucleolar stress response pathway triggers the release of ribosomal proteins from the nucleolus into the nucleoplasm. Moreover, the interaction of NPM and the tumor suppressor ARF (p14 ^ARF^/alternate reading frame) is released. This event inactivates the E3-ubiquitin ligase MDM2/HDM2 (Mouse Double Minute 2), which normally keeps p53 levels low via proteasomal degradation (Figure 2). As a result, the tumor suppressor p53 is stabilized and mediates p53-dependent nucleolar stress responses, such as cell cycle arrest, senescence, DNA damage or apoptosis (Figure 2) [10,11,12]. Of note, specific nucleolar stress pathways can function independently of the p53 status and still have the same or similar outcome, such as apoptosis, cell cycle arrest, senescence and DNA damage [6,12,13]. These p53-independent routes are of particular interest for cancer therapy as p53 reflects the most frequently mutated gene in diverse cancer types [13,14]. Already established clinical chemotherapeutics connected to nucleolar stress are the cytostatics MTX (Methotrexate), 5-FU (5-Fluorouracil) and AcD (ActinomycinD) [7,15]. Drugs of the newer generation, such as the RNA POL I inhibitor CX-5461, are currently tested in clinical trials and can function in a p53-dependent and -independent manner [16,17]. Those inhibitory effects on RNA POL I are depicted in Figure 1.

Importantly, further nucleolar stress responses have recently been uncovered that can again function in a p53-dependent or -independent manner: Nucleolar stress can lead to the activation of autophagy (compare Section 1.2), which either serves as an anti-stress mechanism or pro-death signal [18]. Given the young age of the research field, the precise mechanisms remain to be elucidated. To name a few examples, interfering with RNA POL I function by applying the chemotherapeutics AcD or CX-5461 has been shown to over-activate autophagy in cell culture [17,18,19]. The same holds true when depleting the RNA POL I transcription factors POLR1A or TIF1A (transcription intermediary factor 1A) [19,20]. Likewise, the nucleolar ribosome biogenesis factors (compare Figure 1) NPM and PPAN (Peter Pan), the nucleolar ribosomopathy factor SBDS (Shwachman Bodian Diamond Syndrome) and others have been linked to autophagy [18,19,21,22,23]. However, induction or inhibition of autophagy in response to nucleolar stress has not always been consistent, possibly due to different experimental setups used. 

In the context of pathophysiology, the induction of nucleolar stress can represent a double-edged sword: Applied as a chemotherapeutic, the induction of nucleolar stress serves beneficial purposes to eliminate highly proliferating tumor cells by using anti-proliferative and anti-apoptotic characteristics of the nucleolar stress response [15]. On the other hand, nucleolar stress induction is also connected to cancer, despite the fact that proliferating cells require large amounts of ribosomes. Increased cancer incidence is, for instance, observed for patients with ribosomopathy syndromes [24,25,26,27]. Moreover, increased nucleolar stress seems to be connected to the mechanism of neurodegeneration, in which neurons are lost by apoptosis and show misregulation of autophagy [18]. 

Overall, uncovering the precise underlying mechanisms of the nucleolar stress response in the context of p53 status will be beneficial for enabling further progress for translational applications.

### 1.2. Autophagy

The process of macro-autophagy, commonly referred to as autophagy (“self-eating”), has recently been connected to nucleolar stress [18]. Autophagy is stimulated by various types of cellular stress, most prominently lack of energy and nutrient deprivation. The catabolic process of autophagy is essential for recycling cellular material and for maintaining cellular clearance by eliminating cellular material or pathogens [28,29]. Overall, proper autophagy sustains cell and tissue homeostasis, whereas defects in autophagy cause the accumulation of damaged organelles or protein aggregates. At a certain point, when autophagy is overwhelmed, apoptosis can be induced. As a result, malfunctions give rise to diverse pathological conditions, most prominently neurodegenerative disorders [18]. Moreover, the over-activation of autophagy is considered a resistance mechanism in cancer [30]. Thus, the inhibition of autophagy has been tested as an anti-tumor strategy in clinical trials [31,32]. 

A key characteristic of autophagy is the formation of double-membranous autophagosomes, which engulf the cargo and later fuse with lysosomes (Figure 3). Autophagosomes are built by elongation and closure of membrane precursors, which are termed phagophores [29]. 

The formation of autophagosomes requires the coordinated action of several so-called autophagy-related (ATG) proteins, originally identified in yeast. Some of the ATG proteins are regulated by mTOR and AMPK (AMP-activated protein kinase) signaling. The serin/threonine protein kinases ULK1 (ATG1 homologue) and ULK2 are such examples. Initiation of autophagosome production is mediated by activation of the ULK1-ATG13-FIP200 complex [33]. Additionally, ATG101, which shows no homology to other ATGs, interacts with ULK1 and ATG13 and localizes to the phagophore, where it is essential for the initiation of autophagy [34,35]. Following the activation of ULK1, PtdIns3P formation is stimulated with the help of the proteins Beclin1, PI3KC3 (phosphaditylinositol III kinase class III) and ATG14L [36]. Autophagosome elongation involves two ubiquitin-like conjugation systems composed of the proteins ATG5-ATG12-ATG16L, as well as LC3(ATG8) (light chain 3) [33,36]. The ATG7 protein functions as a key initiation factor and E1-ubiquitin-like ligase that mediates the assembly of the autophagosomal membrane. Moreover, ATG7 and ATG16L1 are involved in the lipidation of unmodified LC3(ATG8), termed LC3-I, which is inserted in the autophagosomal membrane by attachment of a phosphaditylethanolamine (PE) anchor (Figure 3). Lipidated LC3 is known as LC3-II, and the process of lipidation is critical for the establishment, maturation and expansion of autophagosomal membranes [37,38,39]. WIPI1 and WIPI2 (WD-repeat protein interacting with phoshoinositides) are homologous to ATG18 [36]. WIPI(ATG18) localizes to membranes, such as the phagophore, through interaction with PtdIns(3)P and PtdIns(3,5)P_2_ [36,40,41]. WIPI2 recruits the ATG5–ATG12–ATG16L1 complex and mediates LC3 lipidation together with WIPI1 [36]. Moreover, the complex of ATG2, ATG18 and the transmembrane protein ATG9 is essential for tethering the phagophore to the ER [42].

For studying the dynamic process of autophagy, it has become detrimental to perform so-called autophagy flux experiments, in which lysosomal degradation is blocked [43,44,45]. The reason is that autophagosomes are accumulating either (I) due to increased autophagy induction or (II) following reduced turnover, as observed during inhibition of lysosomal function. Thus, flux studies allow an unbiased approach to discriminate between both routes and to properly discriminate between activation of autophagic flux or inhibition [44]. Here, the amount of LC3-II serves as a key readout for examining the status of the autophagic flux.

Note that the process of autophagy can be subdivided into distinct autophagic pathways since selective forms of autophagy also exist. To name a few, mitophagy stands for the selective clearance of mitochondria and aggrephagy for the removal of aggregates [46,47,48]. In principle, ubiquitin-independent as well as -dependent forms of selective autophagy exist [49]. Either so-called autophagy receptors can directly bind to the cargo, or the cargo is earmarked by ubiquitination and then bound by autophagy receptors. The autophagy receptor p62 (SQSTM, sequestosome 1), for instance, recognizes ubiquitinated cargo with its ubiquitin-binding domain and connects it to the autophagosomes by LC3 interaction using an LIR (LC3 interacting region) domain (Figure 3) [50]. Overall, different autophagy receptors mediate the selectivity of the process [51]. The ubiquitination of the cargo is performed by specific E3-ubiquitin-like ligases. For instance, the E3-ubiquitin-like ligase Parkin specifically ubiquitinates damaged mitochondria for mitophagy, whereas mutated Parkin, as found in patients with Parkinson’s disease, results in the accumulation of defective mitochondria and increased apoptosis rate as a consequence of failure in autophagy [52,53]. 

When it comes to the regulation of autophagy, it should be noted that it is governed at RNA and protein level. Several transcription factors drive the expression of core autophagy genes, and also non-coding RNAs are involved in the regulation processes [33]. For instance, the tumor suppressor p53 possesses dual functions in autophagy in a transcription-dependent and -independent manner: Localized to the nucleus, it induces the expression of autophagy-related genes, such as different *ATGs* and *PRKN* (the Parkin gene) [54,55]. In the cytosol and at mitochondria, it affects mitophagy through interaction with Parkin [56,57,58]. Additionally, the transcription factor TFEB (transcription factor EB) drives, amongst others, the expression of *ATG4*, *ATG9B*, *SQSTM1*, *WIPI*, *UVRAG* and *MAP1LC3B* [54,59]. Moreover, forkhead transcription factors, such as DAF-16/FOXO and PHA-4/FOXA, regulate the process of autophagy and have been linked to longevity, as demonstrated in *C. elegans* [60]. Interestingly, high DAF-16 activity was found in intestinal nucleoli, in line with the involvement of nucleolar function in lifespan extension [61]. Accordingly nucleolar size and autophagy play a key role in the process of aging. Whereas enlarged nucleoli are found in aged cells, small nucleoli are key to longevity [18,62,63]. In line, Nesprin-2, a nuclear envelope anchor protein, has recently been found to control nucleolar homeostasis and size, most likely by affecting autophagic degradation of nucleolar factors such as Fibrillarin [64].

Overall, interfering with nucleolar function and the process of ribosome biogenesis has been noticed as an upstream trigger for autophagy. However, the mechanisms have remained largely unknown given the relatively young age of the research field. Dependency on p53 or mTOR signaling has been reported in some studies [18]. As nucleolar stress is tightly coupled to diverse pathophysiological conditions, understanding the precise mechanisms holds great potential for optimizing the therapy situation of diseases linked to nucleolar stress. 

## 2. Transcriptional Control of Autophagy

### 2.1. Transcriptional Control of Autophagy by the p53 Family

As the role of the p53 family in transcriptional control of autophagy has earlier very well been reviewed [54,65], this section only briefly summarizes targets activated by p53 family members (Table 1). Nuclear p53 transactivates various genes connected to autophagy, such as components of the mTOR, AMPK and PI3K pathways, as well as some of the core autophagy machinery [54,66,67]. p53 promotes autophagy by inhibiting mTOR signaling, and it influences the expression of *TSC2*, *REDD1*, *FOXO3*, *β* and *γ* subunits of *AMPK*, *VMP1*, *SESN1* and *SESN2*. Core autophagy genes upregulated by p53 are *ATG2*, *ATG4*, *ATG7*, *ATG10*, *ULK1*, *ULK2* and *UVRAG*. The family member p63 also drives the expression of several ATGs and can compensate for p53 loss. Targets regulated are: *ATG3*, *ATG4*, *ATG5*, *ATG7*, *ATG9*, *ATG10*, *ULK1*, *BECN1* and *MAP1LC3* [54,66,68]. Additionally, p73 has been found to control *ATG5*, *ATG7* and *UVRAG* expression [54,66,69]. Of note, as p53 closely cross-talks with the transcription factors FOXO3 and E2F, the role of p53 itself in autophagy is sometimes hard to discriminate [54].

### 2.2. Transcriptional Control of Autophagy by Factors Connected to Nucleolar Stress and Ribosome Biogenesis

In the following section, manuscripts are briefly summarized, which show a link between nucleolar function and ribosome biogenesis to transcriptional control of autophagy. 

#### 2.2.1. The POL I Transcription Factor TIF1A

The depletion of the RNA POL I transcription factor TIF1A is connected to the induction of nucleolar stress as well as increased autophagy. In a mouse model of Huntington’s disease, conditional knockout of TIF1A resulted in upregulation of the p53-target *PTEN*, as well as over-activation of autophagy (Table 2) [70]. The tumor suppressor PTEN (phosphatase and tensin homolog) functions by inhibiting mTOR signaling, thereby driving autophagy. Note that PTEN can also activate autophagy through the mTOR inhibitor SESN2 (Sestrin-2) in a p53-independent manner, as shown in A549 and HeLa cells, which possess mutated or nonfunctional p53 [71]. 

Overall, the mechanism of TIF1A function might function as an initial pro-survival response to counteract the stress conditions, at least for a certain time window [70]. Autophagy induction after TIF1A knockdown was also observed in human MCF-7 breast cancer cells expressing EGFP-tagged LC3 as well as in HeLa cervical cancer cells. In this setup, autophagy induction by RNA POL I inhibition is independent of p53 but dependent on NPM protein [19].

#### 2.2.2. Inhibition of RNA POL I by CX-5461

The RNA POL I inhibitor CX-5461 has previously been shown to trigger autophagy in U2OS osteosarcoma cells, which occurs via AMPK/mTOR signaling in a p53-dependent manner [18,20]. In HCT116 colorectal cancer cells, the upregulation of the p53 target (Table 1) and mTOR inhibitor *SESN2* [54] is observed, as well as an increase in *MAP1LC3B* and *CCNG2* levels (Table 2) [76]. The effects occur in a p53-dependent manner and result in increased autophagy. Of note, the overexpression of the tumor suppressor CCNG2/Cyclin G_2_ is associated with the activation of autophagic flux and increased levels of ATG5, ATG7 and Beclin [79]. *CCNG2* has been found by microarray analysis to be upregulated by p53; however, *MAP1LC3B* has not been shown earlier to be a p53 target [76]. Additionally, in the p53-expressing cell lines MCF-7 and U2OS, an upregulation of *SESN2*, *MAP1LC3B* and *CCNG2* was found. Interestingly, the inhibition of autophagy by Chloroquine or BafilomycinA1 could sensitize p53-positive HCT116 cells to CX-5461 mediated nucleolar stress but not p53-negative cells [76], thereby suggesting that induction of nucleolar stress along with interference of autophagy are promising strategies in p53-positive cells. Data from our lab showed an increase in the autophagy core regulators *ATG7* and *ATG16L1* upon CX-5461 treatment in HeLa and U2OS cells [22]. As HeLa is considered to be a p53-negative cell line, the effects might as well be p53-independent. However, this should be directly addressed in the future using corresponding p53 knockout cells. Interestingly, *ATG7* levels were also increased in HEK293A cells stably expressing GFP-tagged LC3 after treatment with Chloroquine, which was used to show an increase in autophagic flux in response to CX-5461 (Table 2) [22]. Moreover, we found an increase in *ATG4A*, *ATG5*, *ATG9* and *ULK1* in response to Chloroquine and CX-5461 co-treatment in HEK293A GFP-LC3 cells.

#### 2.2.3. The Nucleolar Acetyltransferase NAT10 

NAT10 (N-acteyltransferase 10) functions as a nucleolar acetyltransferase, which controls ribosome biogenesis. NAT10 itself can be acetylated, and its acetylation status controls the transition from ribosome biogenesis to autophagy [18]. Under normal conditions, NAT10 is acetylated. It then drives ribosome biogenesis, whereas autophagy is inhibited. NAT10 acetylates the RNA POL I transcription factor UBF-1 and activates rRNA transcription. Moreover, it promotes the processing of *18S* rRNA [72]. The inhibition of autophagy is controlled at the transcriptional level: NAT10 overexpression results in decreased expression of the p53-target *REDD1* (Table 1) [54] and *DEPTOR*, thereby inhibiting autophagy. In detail, NAT10 acetylates and inhibits Che-1 (also known as AATF, apoptosis antagonizing transcription factor), a transcription factor of *REDD1* and *DEPTOR*. As REDD1 and DEPTOR are inhibitors of mTOR, autophagy is downregulated in this situation. Of note, transcriptional effects on *REDD1* and *DEPTOR* are observed in p53-positive and -negative HCT116 cells, suggesting p53-independent regulation [72]. In contrast, the knockdown of NAT10 leads to increased *REDD1* and *DEPTOR* levels, thereby upregulating autophagic activity (Table 2) [72]. Importantly, the effect is also observed in p53-negative cells suggesting p53-independent effects on autophagy [72]. Under nutrient deprivation, NAT10 is deacetylated by Sirt1, and then ribosome biogenesis is OFF and autophagy ON. Mechanistically, the inhibition of the transcription factor Che-1 is released and is deacetylated. As a consequence, *REDD1* and *DEPTOR* are induced and mediate increased autophagy [72].

#### 2.2.4. The Mitochondrial Factor MRPL35

Mitochondrial ribosomal proteins (MRPs) are encoded by nuclear DNA and are essential for regulating mitochondrial function. So far, MRPL35 has not been linked to the induction of nucleolar stress. However, it would be interesting to study this aspect, given some similarities observed when interfering with MRPL35 function. MRPL35 protein regulates mitochondrial protein synthesis and OXPHOS assembly [80]. Moreover, MRPL35 overexpression has been found in colorectal cancer and is associated with poor overall survival. MRPL35 depletion increases ROS levels and induces DNA damage, as evidenced by increased γH2AX levels [73]. Loss of MRPL35 stabilizes p53 protein and triggers apoptosis. Autophagy-dependent cell death is induced as the pan-caspase inhibitor z-VAD rescues activation of caspases, but not the cell death per se. In detail, protein levels of ATG5 and the lysosomal protein DRAM1 (DNA-damage regulated autophagy modulator 1) are increased in HCT116 cells and mediate autophagy (Table 2) [73]. Although not demonstrated at the RNA level in this publication, the authors show that the increase in DRAM1 and ATG5 protein depends on p53. Of note, DRAM1 is a direct p53 target, and DRAM1-mediated autophagy is required for apoptosis induction [81]. Thus, it is likely that also, in this case, the transcriptional regulation of autophagy is observed. 

#### 2.2.5. The Ribosomal Protein uL3

Ribosomal protein uL3 mediates rRNA processing, and its levels cause different responses regarding nucleolar stress or autophagy [74]. uL3 protein has been found to be downregulated in colon tissues. uL3 depletion in HCT116 p53^−/−^ cells mediates chemoresistance, affects RNA maturation and triggers nucleolar stress and increases autophagy. *ATG13*, *ATG101* and *ULK1* RNAs are upregulated in uL3-depleted cells and also *TFEB* expression is increased in p53 knockout cells. In contrast, uL3 overexpression has the opposite effect and decreases levels of *ATG13*, *ATG101*, *ULK1* and *TFEB* and inhibits autophagy independently of p53 (Table 2) [74]. Of note, the transcription factor TFEB is known to induce further autophagy target genes such as *ATG4*, *ATG9*, *MAP1LC3B*, *SQSTM1*, *UVRAG* and *WIPI* [54,59], some of which have already been linked to interfering with ribosome biogenesis. uL3 can function as a sensor of nucleolar stress. Its levels increase in response to chemotherapeutics, such as AcD, independently of p53 status. uL3 overexpression sensitizes cells to chemotherapeutics independently of p53. uL3 overexpression induces nucleolar stress, RNA POL I transcription is inhibited, and *47S* rRNA levels decrease. Moreover, uL3 overexpression decreases levels of *ATG13*, *ATG101*, *ULK1* and *TFEB* independently of p53. As a consequence, the cell cycle is arrested in G_1_ phase, apoptosis is induced and autophagy is inhibited [74].

#### 2.2.6. The Nucleolar Factor Nopp140 

Nopp140 is a nucleolar phosphoprotein whose loss triggers nucleolar stress and apoptosis, and it has further been linked to the induction of autophagy [18]. Interestingly, it is functionally and structurally related to Treacle, a factor connected to ribosomopathies [82,83]. Nopp140 depletion in *Drosophila* activates the MAPK/JNK pathway and induces (likely via dFOXO) the autophagy genes *ATG1*, *ATG18.1* and *ATG8a* (Table 2) [75]. It is known that FoxOs are major regulators of *ATG* transcription. FoxO can directly bind to the LC3 promoter [54]. Thus, transcription-dependent autophagy can likely occur via JNK and FoxOs, thereby activating autophagy.

#### 2.2.7. Depletion of LAS1, PELP1 or NOP1

Laio et al. have performed gene expression analysis and studied effects induced by nucleolar stress in response to p53 stabilization. The authors show that interfering with ribosome biogenesis triggers a pro-autophagy program already at the transcriptional level. Nucleolar stress is induced via the depletion of ribosome processing factors LAS1L (LAS1-like ribosome biogenesis factor), PELP1 (Proline, Glutamic acid and Leucine-rich protein 1) or NOP2 nucleolar protein in HCT116 cells. LAS1L functions as an endoribonuclease involved in *28S* RNA processing and synthesis of the 60S ribosomal subunit. PELP1 and NOP2 also participate in 60S ribosome biogenesis. LAS1L, PELP1 or NOP2 depletion leads to an increase in the autophagy regulators *SESN2*, *CCNG2* and *MAP1LC3B* (Table 2) [76]. Moreover, *WIPI1* was found to be upregulated in an initial microarray after LASL1 depletion. The expression of SESN2, *CCNG2* and *MAP1LC3B* is p53 dependent, as shown in HCT116 p53^−/−^ cells depleted for LASL1. Of note, *SESN2* already represents a known p53 target [54], whereas *CCNG2* has been found in microarrays to be regulated by p53. In contrast, *MAP1LC3B* or *WIPI1* have not been shown earlier to be p53 targets [76].

#### 2.2.8. The Nucleolar Factor NOP53

NOP53, also known as PICT1 (Protein Interacting with C-Terminus) or GLTSCR2 (Glioma Tumor Suppressor Candidate Region Gene 2), functions as a ribosome biogenesis factor, which binds and stabilizes p53 and also reduces NPM levels via proteasomal degradation. NOP53 overexpression in LN18 glioblastoma cells activates *ZKSCAN* expression and, with it, reduces *MAP1LC3B* levels in a ZKSCAN3-dependent manner [77]. In line, ZKSCAN3 represses autophagy and affects transcription of, e.g., *MAP1LC3B* [54,84]. Additionally, *ATG7* and *ATG12* levels are downregulated, however, in a ZKSCAN3-independent manner. Overall, NOP53 overexpression inhibits autophagy, and the autophagy regulation is neither dependent on p53 nor NPM. Interestingly, NOP53 overexpression reduces *ATG7* and *ATG12* via its interaction with Histone H3 [77]. In contrast, NOP53 knockdown increases levels of *ATG5*, *ATG12* and *MAP1LC3B*.

#### 2.2.9. Depletion of Nucleolar Factors PPAN, NPM, SBDS, PES1 or UBF-1

We have previously shown that induction of nucleolar stress by independent strategies induces changes in core autophagy machinery at the RNA level. We depleted key factors involved in the production of ribosomes: PPAN and PES1 (Pescadillo 1), which mediate the processing of the 60S precursor and NPM that functions as endoribonuclease and interacts with the ribosomopathy factor SBDS. Moreover, we depleted UBF-1, a transcription factor of RNA POL I [22]. Interfering with PPAN, NPM, PES1, SBDS and UBF-1 all over-activated the expression of *ATG7* as well as *ATG16L1* levels in HeLa and U2OS cells. With the exception of PPAN knockdown, also levels of *ATG5* were increased in all conditions tested (Table 2) [22]. Moreover, further factors were upregulated in HeLa cells, such as *ATG4A*, *ATG9* and *ULK1* [22]: *ATG4A* is increased after PPAN, NPM, PES1 or SBDS knockdown and *ATG9* after the knockdown of NPM, PES1 and SBDS. *ULK1* is increased after the knockdown of PES1, SBDS and UBF-1. Together this shows that interference with ribosome biogenesis provides mRNA of factors implicated in the early steps of autophagy. An increase in autophagy flux has so far been demonstrated for PPAN knockdown in HeLa cells and HEK293A GFP-LC3 cells [23]. SBDS has been shown to increase autophagy in leukocytes derived from patients with ribosomopathy SBDS and in epithelial cells, it has been shown that the effect is independent of mTOR or p53 [18,21]. In contrast, NPM depletion counteracts autophagy in TIF1A-depleted MCF-7 cells expressing EGFP-tagged LC3 [19].

#### 2.2.10. Depletion of the Nucleolar Factor DKC1

DKC1 (Dyskerin) is a nucleolar factor that functions as pseudouridine synthase with multiple roles [85]. DKC1 binds H/ACA box motifs present in snoRNAs as well as to the telomerase component TERC (Telomerase RNA Component), which is responsible for telomere maintenance. Besides others, DKC1 drives ribosome biogenesis by mediating rRNA pseudouridylation and sustains telomerase function. DKC1 is mutated in the multisystemic disease X-linked dyskeratosis congenita (X-DC), which affects the skin, causes bone marrow failure and predisposes patients to cancer. In line with its functions, mutations in DKC1 mediate symptoms of telomeropathies as well as ribosomopathies. 

To monitor early and thus telomere-independent effects, an inducible depletion of DKC1 has been used [78]. In colorectal RKO and kidney HEK293T cells an increase in autophagic markers such as ATG5, ATG12, LC3 and BECN1 was observed and went together with an increase in GFP-LC3 puncta and autophagic flux as well as impaired mTOR signaling. Upregulation of *ATG5* and *ATG12* was detected at the mRNA level [78], suggesting activation of autophagy already at the transcriptional level. Also, ATF4 was increased in RKO and HEK293T cells [78], which can, in principle, function as a transcription factor for *ATG5* and *MAP1LC3B* [54,86].

## 3. Concluding Remarks

Overall, the regulation of *ATG* expression has been found by multiple independent approaches in different settings related to interference with ribosomal function (Table 3). 

For instance, *ULK1*, or its homolog *ATG1*, are upregulated in response to uL3 depletion and Nopp140 knockdown, respectively [74,75]. *ATG5* upregulation is found via MRPL35 depletion and is p53-dependent [73]. *ATG5* upregulation is likewise found after UBF1, PES1, NPM and SBDS knockdown [22] or by DKC1 depletion [78]. *ATG7*, is upregulated via the depletion of several nucleolar factors, such as PPAN, NPM, SBDS, PES1 and UBF-1 as well as by CX-5461 treatment [22] or is downregulated by NOP53 overexpression [77]. *ATG12* is upregulated in DKC1-depleted cells [78] or downregulated after NOP53 overexpression [77]. In addition, also *MAP1LC3* expression is affected by different approaches (Table 3): LAS1L, PELP1 or NOP2 depletion or CX-5461 treatment increases *MAP1LC3B* levels [76] and NOP53 overexpression decreases *MAP1LC3B* levels [77], latter effect is ZKSCAN3-dependent. 

Overall, interference with nucleolar function and ribosome biogenesis seems to be tightly coupled to transcriptional changes, thereby affecting the process of autophagy. Also, here, p53-dependent and -independent mechanisms have been identified. This suggests that perturbation of ribosome biogenesis might provide factors of the core autophagy machinery already at a transcriptional level to drive autophagic flux, at least as an initial response to nucleolar stress. 

As the research area around nucleolar stress and autophagy is relatively new, many mechanisms remain to be elucidated in future studies. It remains to be determined whether and which common transcriptional patterns exist and how exactly p53-independent effects are regulated. For instance, the expression of core autophagy genes, such as diverse *ATG*s, *MAP1LC3*, *ULK*, *SQSTM* or *WIPI*, previously also linked to nucleolar stress, can be regulated by multiple other transcription factors than the p53 family: ATF4 and ATF5, β-Catenin, C/EBPβ, CHOP, E2F1, FOXO1 and FOXO3, GATA1, HIF1, JUN, NF-κB, SOX2, SREBP2, STAT1 and STAT3, TFEB and ZKSCAN3 [54]. 

Given the tight connection between apoptosis and autophagy, many questions arise as to when and how the transition from autophagy to apoptosis occurs in the context of nucleolar stress. A better understanding of those mechanisms could be exploited in future therapies against diseases connected to nucleolar stress. 

## Figures and Tables

**Figure 1 cells-12-02071-f001:**
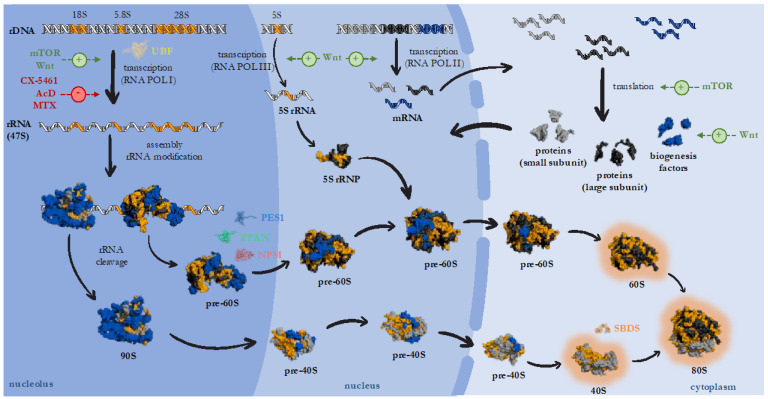
Mammalian ribosome biogenesis, under control of mTOR and Wnt signaling. rDNA is transcribed with help of the transcription factor UBF-1 (yellow) by RNA POL I. The ribosome biogenesis factors PPAN (green), PES1 (blue) and NPM (red) function in biogenesis of the large ribosomal subunit. SBDS (orange) is implicated in late processing steps in the cytoplasm. The positive role of mTOR and Wnt signaling is depicted in green (+), the inhibitory role of the chemotherapeutics CX-5461, AcD (ActinomycinD) and MTX (Methotrexat) is depicted in red (−). The protein models were generated using the phyton-based open source PyMol (Schrodinger, LLC, New York, NY, USA).

**Figure 2 cells-12-02071-f002:**
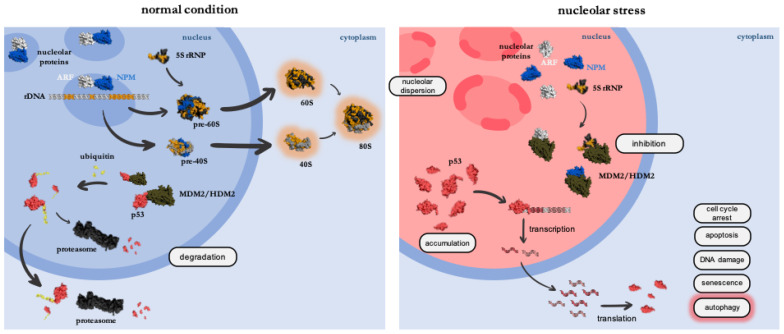
The nucleolar stress response. Under normal conditions (left side), NPM (blue) and ARF (white) interact within nucleoli. Thus, p53 (red) levels are kept low by proteasomal (black) degradation involving MDM2 (brown)-mediated ubiquitination (yellow) of p53. As a consequence of nucleolar stress (right side), nucleolar and ribosomal proteins are released. The 5S rRNP complex binds and impairs MDM2 function. Moreover, ARF is released from NPM into the nucleoplasm and inhibits MDM2. In turn, p53 accumulates, and p53-mediated effects are propagated (circled boxes). Autophagy (red) represents the most recently identified response. The protein models were generated using the phyton-based open source PyMol (Schrodinger, LLC, New York, NY, USA).

**Figure 3 cells-12-02071-f003:**
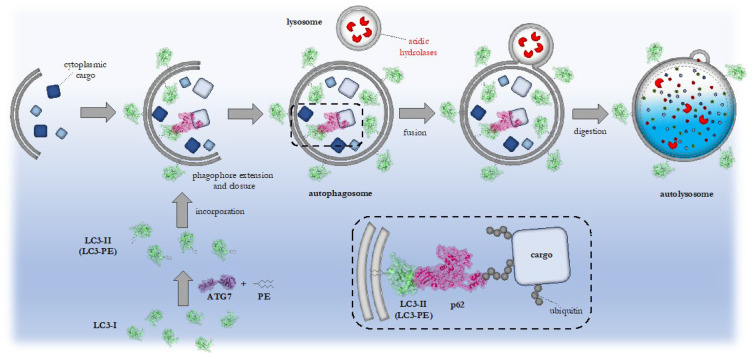
A simplified model of autophagy. The double-membranous phagophore (gray) contains LC3-II protein (green) and forms around the bulk cargo (blue and gray boxes). The phagophore matures into the autophagosome and subsequently fuses with the lysosome containing hydrolases to form the autolysosome, in which the cellular material is degraded by acidic hydrolases (red). LC3 is lipidated with the help of ATG7 (purple). The autophagy receptor p62 (pink) functions in ubiquitin-dependent selective autophagy by interacting with LC3-II and the ubiquitinated cargo (gray). The protein models were generated using the phyton-based open source PyMol (Schrodinger, LLC, New York, NY, USA).

**Table 1 cells-12-02071-t001:** Autophagy targets transactivated by the p53 family (compare with the review [54]).

p53 Family Member	Regulated Genes	Reference
**p53**	*ATG2*, *ATG4*, *ATG7*, *ATG10*, *ULK1*, *ULK2*, *UVRAG*, *TSC2*, *REDD1*, *SESN1*, *SESN2*, *FOXO3*, *VMP1*, *b AMPK*, *g AMPK*	[66,67]
**p63**	*ATG3*, *ATG4*, *ATG5*, *ATG7*, *ATG9*, *ATG10*, *ULK1*, *BECN1*, *MAP1LC3*	[66,68]
**p73**	*ATG5*, *ATG7*, *UVRAG*	[66,69]

**Table 2 cells-12-02071-t002:** Factors and drugs connected to ribosome biogenesis and transcriptional control of autophagy. +, up-regulation/activation; −, down-regulation/inhibition; n.d., not determined. Publications are listed in chronological order.

Manipulation	Target	Autophagy	Dependency	Reference
**TIF1A depletion**	*PTEN* +	+	p53-dependent	[70]
**NAT10 knockdown**	*REDD1* + *DEPTOR* +	+	p53-independent	[72]
**MRPL35 depletion**	*ATG5* + *DRAM* +	+	p53-dependent	[73]
**uL3 knockdown**	*ATG13* + *ATG101* + *ULK1* + *TFEB* +	+	p53-independent	[74]
**Nopp140 depletion**	*ATG1* + *ATG8a* + *ATG18.1* +	+	n.d.	[75]
**LAS1L depletion;** **CX-5461 treatment**	*SESN2* + *MAP1LC3B* + *CCNG2* +	+	p53-dependent	[76]
**PELP1 depletion;** **NOP2 depletion**	*SESN2* + *MAP1LC3B* + *CCNG2* +	+	n.d.	[76]
**NOP53 overexpression**	*ATG7* − *ATG12* − *MAP1LC3B* − *ZKSCAN3* +	−	ZKSCAN3-independent (*ATG7*, *ATG12*) ZKSCAN3-dependent (*MAP1LC3B*)	[77]
**UBF-1 knockdown;** **SBDS knockdown;** **PPAN knockdown;** **NPM knockdown;** **PES1 knockdown;** **CX-5461 treatment**	*ATG7* + *ATG16L1* +	+ (CX-5461; PPAN knockdown)	n.d.	[22,23]
**UBF-1 knockdown;** **SBDS knockdown;** **NPM knockdown;** **PES1 knockdown**	*ATG5* +	n.d.	n.d.	[22]
**DKC1 depletion**	*ATG5* + *ATG12* +	+	n.d.	[78]

**Table 3 cells-12-02071-t003:** Autophagy core machinery regulated by various nucleolar factors or CX-5461 as indicated. +, up-regulation/ activation; −, down-regulation/ inhibition.

Target	Manipulation	Reference
*ULK1/ATG1 +*	uL3 knockdown;	[74]
	Nopp140 depletion	[75]
*ATG5 +*	MRPL35 depletion;	[73]
	UBF-1 knockdown;	[22]
	SBDS knockdown;	
	NPM knockdown;	
	PES1 knockdown	
	DKC1 depletion	[78]
*ATG7 +*	UBF-1 knockdown;	[22]
	SBDS knockdown;	
	PPAN knockdown;	
	NPM knockdown;	
	PES1 knockdown;	
	CX-5461 treatment	
*ATG7* −	NOP53 overexpression	[77]
*ATG12 +*	DKC1 depletion	[78]
*ATG12* −	NOP53 overexpression	[77]
*MAP1LC3B +*	LAS1L depletion;	[76]
	PELP1 depletion;	
	NOP2 depletion;	
	CX-5461 treatment	
*MAP1LC3B* **−**	NOP53 overexpression	[77]

## Data Availability

No new data were created here.

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
