# Peer review of "An Update on Nucleolar Stress: The Transcriptional Control of Autophagy"

_cells, 2023, doi:10.3390/cells12162071_

Round 1

Reviewer 1 Report

In this review, the author reports on recent evidence implicating nucleolar stress in the regulation of autophagy.  In particular, the author focuses on p53-dependent and -independent mechanisms by which nucleolar stress modulates autophagy at the transcriptional level. The topic of the article is interesting. The review is generally very well written and organised, but there is room for improvement. 

Specific comments are summarised below:

- Some information about autophagy pathways is not sufficiently precise. For example, it is noted that cargo ubiquitination by specific E3-ubiquitin-like ligases is a common mechanism of both general autophagy and selective forms of autophagy. However, ubiquitin-independent forms of autophagy are also known (Khaminets et al., Trends in Cell Biology, 2016, Vol. 26).

- In addition to TFEB and p53, other transcription factors, such as the forkhead transcription factors, DAF-16/FOXO and PHA-4/FOXA, are known to upregulate certain autophagy genes. This information should also be included in the article.

- As an expert in the field, the author has published other reviews on the subject. It is therefore plausible that there is some repetition of previous articles. For example, Figures 1 and 2 are similar in concept and design to those published in Front. Cell. Neurosci., 2019, vol. 13, 156.

The new information presented in this article focuses on the transcriptional control of autophagy in relation to nucleolar function and ribosome biogenesis. In this context, the author summarizes recent observations showing that depletion of nucleolar factors upregulates autophagy genes. Therefore, this article would be more impactful if the findings could be synthesised into a more coherent picture. This will also help to maximise the potential impact of the article.

- A recent study has shown that nesprins play a key role in determining nucleolar size and function and thus in controlling nucleolar homeostasis, most likely through autophagy. As a consequence, nesprins contribute to somatic longevity and germline immortality (Papandreou et al., Nature Aging 2022, vol 3, pp 34–46). The author may wish to include this study where relevant.

Display items

The article currently contains 2 Figures and 2 Tables, which overall add to the review. However, one or two additional figures giving molecular details of the mechanisms discussed would enhance the value of the article.

Textual points

-Some syntax/language errors throughout the text need to be corrected.

The review is generally very well written. However, some syntax/language errors need to be corrected.

Author Response

Please see attached PDF with response

Reviewer 2 Report

This paper cold give a contribution in its field of interest, even if the subject has been largely stressed by the scientific community, and for this reason it lacks of novelty. The author very well explained the mechanisms at the basis of the nucleolar stress, and also the adaptive response of the cells to this condition. Nevertheless, some issues must be addressed to meet the journal standards.

1. In lane 55, the author missed informations about the cell morphological changes promoted by the nucleolar stress, as suggested by the folowing papers:

Yang K, et al. A redox mechanism underlying nucleolar stress sensing by nucleophosmin. Nat Commun. 2016

Yang K, Yang J, Yi J. Nucleolar Stress: hallmarks, sensing mechanism and diseases. Cell Stress. 2018 PMID: 31225478.

2. As reported in literature, there are some other p-53-independent response to the nucleolar stress, not only leading to the autophagy. The authors should report some informations on this. This reviewer suggest the following papers:

James A, et al. Nucleolar stress with and without p53. Nucleus. 2014 PMID: 25482194

 In this review is reported an interesting study on a p-53 independent autophagy activation in nucleolar stress condition:

Boglev Y, et al. Autophagy induction is a Tor- and Tp53-independent cell survival response in a zebrafish model of disrupted ribosome biogenesis. PLoS Genet. 2013

3. The author should improve the Figure 1, which is very similar to the one already published in:

Pfister AS. Emerging Role of the Nucleolar Stress Response in Autophagy. Front Cell Neurosci. 2019 Apr 30 PMID: 31114481

In particular, the author should represent the mechanisms underlying the nucleolar stress response, also reporting in picture the molecular effectors involved in the latters (rRNA biogenesis maturation and regulation by the mTOR and Wnt signalling; the effect of some drugs on these mechanisms;  the role of MDM2; etc...). Moreover, the author should report also the main effectors involved in the nucleolar stress induced autophagy (NPM (Nucleophosmin), PPAN (Peter Pan), or the nucleolar ribosomopathy factor SBDS (Shwachman Bodian Diamond Syndrome). The p53-independent development of the flow-chart is completely missed; the authors should mention into the picture alternative p53-independent mechanisms.

4. In Lane 118, the citation referred to the sentence: “autophagy is considered a resistance mechanism in cancer” is missing. This reviewer suggest to cite the following papers:

Mele L, et al. β2-AR blockade potentiates MEK1/2 inhibitor effect on HNSCC by regulating the Nrf2-mediated defense mechanism. Cell Death Dis. 2020  PMID: 33051434;

Xie X, et al. Coordinate autophagy and mTOR pathway inhibition enhances cell death in melanoma. PLoS One. 2013; PMID: 23383069.

5. The figure 2 is not acceptable because is practically the same already published in:

Pfister AS. Emerging Role of the Nucleolar Stress Response in Autophagy. Front Cell Neurosci. 2019 Apr 30 PMID: 31114481

This reviewer suggests to ameliorate the picture, reporting the autophagy regulation machinery, especially referring to the molecular effectors directly or indirectly regulated by the p53 family (and eventually the p-53-independent signalling).

The quality of English language is acceptable.

Author Response

Please see attached PDF with response

Reviewer 3 Report

Title: An update on nucleolar stress: The transcriptional control of 2 autophagy by Pfister et al.

I found, that the topic is original and relevant in the field.

I found the conclusion to be in line with the evidence and arguments presented.

The references are well-updated.

The manuscript is interesting, however, it can be improved and strengthened by addressing the following comments -

1. The author must provide a reference within 5 years (2019-2023).

2. A significant paper, PMID: 37048159, Autophagy activation by PTEN in A549 cells that is independent of p53, is missing.

3. The last paragraph of the " Concluding Remarks" could be more extended.

Minor editing of the English language required

Author Response

Please see attached PDF with response

Round 2

Reviewer 3 Report

The revised manuscript has shown considerable enhancements and has effectively addressed all of my concerns. Based on its current state, I recommend accepting this manuscript for publication.

Minor editing of the English language is required.